# Introducing Adam’s Scale of Posterior Stroke (ASPOS): A Novel Validated Tool to Assess and Predict Posterior Circulation Strokes

**DOI:** 10.3390/brainsci11040424

**Published:** 2021-03-26

**Authors:** Adam Wiśniewski, Karolina Filipska, Katarzyna Piec, Filip Jaskólski, Robert Ślusarz

**Affiliations:** 1Department of Neurology, Faculty of Medicine, Collegium Medicum in Bydgoszcz, Nicolaus Copernicus University in Toruń, 85-094 Bydgoszcz, Poland; piec.kasia@gmail.com (K.P.); filipjask@gmail.com (F.J.); 2Department of Neurological and Neurosurgical Nursing, Faculty of Health Sciences, Collegium Medicum in Bydgoszcz, Nicolaus Copernicus University in Toruń, 85-821 Bydgoszcz, Poland; karolinafilipskakf@gmail.com (K.F.); robert_slu_cmumk@wp.pl (R.Ś.)

**Keywords:** stroke, clinimetrics, posterior circulation stroke, reliability, validity, prediction, neurological score, stroke assessment

## Abstract

Background: Assessing the severity of posterior circulation strokes, due to the variety of symptoms, is a significant clinical problem. Current clinimetric scales show lower accuracy in the measurement of posterior stroke severity, compared with that of anterior strokes. The aim of the study was to design a validated tool, termed Adam’s Scale of Posterior Stroke (ASPOS), for better assessment and prediction of posterior stroke. Methods: This prospective, observational study involved 126 posterior circulation ischemic stroke subjects. Four researchers, previously trained in ASPOS, randomized the stroke severity using a novel tool and other appropriate stroke scales (The National Institute of Health Stroke Scale—NIHSS, modified Rankin Scale—mRS, Glasgow Coma Scale, Barthel Index, or Israeli Vertebrobasilar Stroke Scale—IVBSS) to assess the psychometric properties, reliability, and validity of ASPOS and investigate its predictive value. Results: ASPOS reached a Cronbach’s alpha coefficient of 0.7449, indicating good internal consistency. The Bland–Altman analysis showed a good coefficient of repeatability (CR) of 0.46, a 95% confidence interval (CI) of 0.41–0.53, and excellent intraclass correlation coefficients or weighted kappa values (>0.90), reflecting high reliability and reproducibility. Highly significant correlations with other scales confirmed the construct and predictive validity of ASPOS. A total ASPOS score of three points indicated a significantly increased probability of severe stroke based on the NIHSS, compared to a total ASPOS of 1–2 points (odds ratio (OR) 141; 95% CI: 6.72–2977.66; *p* = 0.0014). Conclusions: We developed a novel, valid, and reliable tool to assess posterior circulation strokes. This can contribute to a more comprehensive estimation of posterior stroke and, additionally, due to its predictive properties, it can be used to more accurately select candidates for specific treatments.

## 1. Introduction

Posterior circulation strokes account for 20–40% of all ischemic strokes [1]. Compared to anterior circulation strokes, this type is characterized by a greater complexity of clinical symptoms, greater unpredictability, and clinical variability. Ischemia of various areas that are supplied by the posterior circulation, including the occipital region, brainstem, and cerebellum, leads to diverse clinical manifestations that often pose a great diagnostic challenge for physicians [2]. The National Institutes of Health Stroke Scale (NIHSS) is the scale most widely used to assess stroke severity. It is much more accurate when capturing the course of a stroke in anterior circulation as it does not include clinical elements typical of posterior circulation, such as nystagmus or gait disturbances, leading to the underestimation of stroke severity in these cases [3,4,5,6]. There are still doubts regarding whether the NIHSS can be used for posterior strokes, as reflected, for example, in the qualification of thrombectomy. In patients with anterior stroke, there is a general agreement regarding the significant risk of large intracerebral vessel occlusion (i.e., 6 points on the NIHSS scale). However, there is no such limit and consensus for posterior stroke [7]. Moreover, assessing the indications for extended vascular diagnostics is normally left to the physician’s discretion. The heterogeneity and complexity of clinical symptoms are the main reasons for the lack of a clinimetric tool dedicated to this group of strokes. Numerous reports have signaled the need to unify and standardize these stroke groups. Unfortunately, only a few authors have presented the development of such a dedicated scale (the Israeli Vertebrobasilar Stroke Scale—IVBSS) [8], based on a small population of patients, or have attempted to make an extended version of NIHSS [9]. Ultimately, however, they have not found practical and widespread application, mainly due to the lack of attempts aimed at validation, modification, or improvement to create a common, recognized, and accepted clinimetric instrument.

The aim of the current study was to develop and validate a consolidated, reliable, and reproducible clinimetric tool (Adam’s Scale of Posterior Stroke—ASPOS) dedicated to exclusively assessing the severity of stroke in posterior circulation with additional predictive properties.

## 2. Materials and Methods

### 2.1. Study Design and Participants

This prospective and observational study was conducted from November 2019 to September 2020 in the Department of Neurology at University Hospital No. 1 in Bydgoszcz, Poland. This study included 126 patients who met the clinical and radiological criteria for the diagnosis of ischemic stroke in posterior circulation (onset within 24 h of admission).

Both clinical and functional condition were assessed within 24 h after the onset of stroke using the NIHSS, Glasgow Coma Scale (GCS), IVBSS, Barthel Index, and modified Rankin Scale (mRS), all of which are relevant and common instruments for this purpose. Four researchers, including two physicians, one stroke research nurse, and a physiotherapist, performed the stroke subject evaluation; each had several years of experience working in the intensive stroke unit and had NIHSS assessment certification.

On the first day of the stroke, three randomly selected investigators assessed each patient using ASPOS for estimation of the inter-rater reliability. The differences in the assessment did not exceed 2 h. One researcher, randomly selected each time, assessed the clinical and functional condition of stroke subjects with other available scales on the first day after the stroke to estimate the construct validity, and predictive validity was estimated on the 90th day after the stroke. Three hours after the ASPOS evaluation, one of the three previously randomly selected investigators was also selected at random for re-assessment by ASPOS (test–retest) to estimate the intra-rater reliability. The differences in the sum values of the ASPOS between two randomly selected researchers were used to assess the repeatability of the tool.

The following exclusion criteria were used: lack of consent for the patient to participate in the study or inability to express it consciously (e.g., stroke with quantitative and qualitative disturbances of consciousness). Furthermore, subjects underwent reperfusion therapy (thrombolysis and/or thrombectomy) due to the tendency for rapid and significant fluctuations in stroke severity.

### 2.2. ASPOS

ASPOS (Table 1) consists of seven items, each scored from 0 to 2 or 3 points. The maximum number of points that can be achieved is 19. In case of doubts as to the severity of a given symptom or several options to choose from within one item, the option corresponding to the higher score was indicated. Researchers who had undergone NIHSS training also completed ASPOS training based on repeated clinical examinations of all items. The general principles of evaluation did not differ significantly from those applied in NIHSS (e.g., when it was not possible to evaluate a given parameter for reasons other than stroke, it was not added to the total score) [10,11,12]. First, we have screened subjects with posterior stroke hospitalized in our Department from 2017 to 2018 to identify and characterize the most common neurological symptoms. Then, we categorized individual symptoms and created initial items in different configurations, giving different scores on a scale from 0 to max. 3 points for each item. Ultimately, we created 6 different preliminary versions with 6 to even 9 items with different scores. All versions were refilled during the study of each patient. After completing the study, each preliminary version was analyzed in terms of psychometrical parameters, reliability and constructive and predictive accuracy. The final version was the one that achieved the highest design, structural and clinical properties.

### 2.3. Ethical Statement

The study protocol was approved by the Bioethics Committee of the Nicolaus Copernicus University in Torun at Collegium Medicum of Ludwik Rydygier in Bydgoszcz (KB number 733/2019). All subjects read the study protocol and signed informed consent to participate in the study. The study was conducted according to the Declaration of Helsinki regarding research on humans.

### 2.4. Statistical Evaluation Methods

The statistical analysis of the collected data was performed with STATISTICA version 13.1 (Dell Technologies, Round Rock, TX, USA). Due to the incompatibility of feature distribution with normal distribution (Shapiro–Wilk test), nonparametric tests were used, i.e., the Mann–Whitney U test (assessment of the predictive relation between ASPOS and NIHSS), Spearman’s rank correlation test (evaluation of construct and predictive validity), intraclass correlation coefficient, and weighted Cohen’s kappa (evaluation of inter-rater and intra-rater reliability). The psychometric properties of the tool and repeatability were assessed by Cronbach’s alpha coefficient and Bland–Altman analysis, respectively. Logistic regression model and Kruskal–Wallis statistics were performed for the evaluation of the predictive values of ASPOS. The significance level of *p* < 0.05 was considered as the threshold for statistical significance.

## 3. Results

The general characteristics of the studied population are shown in Table 2. The median total ASPOS score was 2 points (min.–max. 1–11). A Cronbach’s alpha coefficient of 0.74 was reached, showing good internal consistency. Statistics for individual items are presented in Table 3. Only one item (eyes) did not reach statistical significance in correlation with the others, and its removal increased the consistency of the entire scale. The remaining items obtained a high correlation value in relation to the others (R > 0.50), and their removal from the score lowered the overall internal consistency of the tool.

The results of the inter-rater and intra-rater reliability are shown in Table 4. All items achieved excellent intraclass correlation coefficient (ICC > 0.9) and weighted kappa (κ > 0.9) scores. The Bland–Altman analysis (Figure 1) showed a good coefficient of repeatability (CR 0.46; 95% CI: 0.41–0.53) and a narrow limit of agreements (lower (−0.44; 95% CI: −0.51 to −0.37) and upper limits (0.49; 95% CI: 0.41–0.56)), underlining the accuracy of our device. Only seven pairs of compared scores (5.5%) were outside of the range set by the limits of agreement, but the maximum difference in total score between investigators was one point.

We reported a high correlation of ASPOS with the initial NIHSS (R = 0.87, *p* < 0.0001), Barthel Index (R = −0.92, *p* < 0.0001), and mRS (R = 0.87, *p* < 0.0001), indicating good construct validity (Figure 2A–C). Moreover, a moderate but significant correlation with GCS (R = −0.47, *p* < 0.0001) and a high correlation with IVBSS (R = 0.91, *p* < 0.0001) were found. On the 90th day after the stroke, we revealed high correlations with NIHSS (R = 0.86, *p* < 0.0001), Barthel Index (R = −0.91, *p* < 0.0001), and mRS (R = 0.86, *p* < 0.0001), indicating the high predictive validity of the tool (Figure 2D–F).

In Figure 2 (above), significant correlations between ASPOS and the National Institutes of Health Stroke Scale, the Barthel Index, and the modified Rankin Scale (mRS) on the first day of the stroke (Figure 2A–C) emphasized the high construct validity of ASPOS. Significant correlations between ASPOS and theNational Institutes of Health Stroke Scale, the Barthel Index and the modified Rankin Scale on the 90th day after the stroke (Figure 2D–F) confirmed the high predictive validity of ASPOS. The places where the scales show the most significant correlation were emphasized with different intensity of colors—the degree of correlation varies with the warmer color—from blue, through yellow to red, which symbolizes the point of the highest correlation.

Using an NIHSS value of 6 as the cutoff for stroke severity and qualifying for a thrombectomy, we noted that, in the group with a total ASPOS score of 1 or 2 (*n* = 86, 68.2%), no case of NIHSS with at least six points was found. In the group with a total ASPOS score of three points (*n* = 9; 7.2%), we reported five cases with an NIHSS total score of six points or more. In the ASPOS group with 4–11 points (*n* = 31; 24.6%), 26 cases of NIHSS had six points or more. The Kruskal–Wallis test showed a significant difference between the above groups (H = 78.52; *p* < 0.0001) (Figure 3). Logistic regression showed that a total ASPOS score of three points indicated a significantly increased probability of a severe stroke based on NIHSS, compared with a total ASPOS of 1–2 points (OR = 141; 95% CI: 6.72–2977.66; *p* = 0.0014).

## 4. Discussion

We developed a novel diagnostic tool, guided by three main assumptions, that may be useful in the assessment and prediction of posterior circulation stroke.

First, we aimed for an ASPOS that would be simple, easy to complete, and understandable, such that, like NIHSS, it could be used not only by longtime neurologists but by other members of the stroke team, such as nurses or physical therapists. Limiting the components of the scale to the necessary minimum, a clear definition for individual scale components, clear rules for scoring the severity of individual items, and adopting general rules for assessment similar to those in NIHSS made it possible to achieve high reproducibility between examiners, which was reflected in the excellent ICC and κ values. Moreover, a high rate of repeatability was proven by the Bland–Altman analysis [13,14]. This confirmed the stability and reliability of ASPOS.

Secondly, the scale was designed to meet the requirements of the International Quality of Life Assessment Project to obtain appropriate psychometric parameters [15]. Despite the variety in signs of posterior circulation stoke, it was possible to design a scale that characterizes internal consistency and covers all the most significant clinical symptoms. The scalability and homogeneity of ASPOS was confirmed on the basis of the requirement that the Cronbach’s alpha coefficient be above 0.7 (according to Nunnelly’s principle) [16], indicating that all items measured the same attribute [17]. Moreover, the discriminatory power of individual items of cases exceeded 0.5 in the vast majority (as required, they should take values higher than 0.3) [18]. Only one component of ASPOS (eyes) differed from the others, not reaching a sufficient discriminant power. However, the final version of the scale was a compromise. Attempts to separate this component into several smaller fractions (e.g., visual field, eye movement, or nystagmus) did not increase the discriminatory power against other parameters, decreasing the index of internal consistency, increasing the complexity of the scale, and reducing its reliability due to the lower repeatability index. Therefore, from the point of view of the scale’s simplicity and its psychometric properties, which were our top priority, the best option was to maintain a more complex item. On the other hand, the complete removal of this component from the scale, to achieve higher internal consistency, was not considered due to the special importance of this group of clinical symptoms, specifically for posterior strokes. Nevertheless, it is worth mentioning that ASPOS obtained better psychometric parameters, including alpha coefficient and discriminating power of particular components, than IVBSS scale, as validated by our team [19].

Finally, we devised a valid tool to accurately estimate the stroke severity based on the high correlations with other widely recognized scales used for this purpose. Significant dependencies on the first and 90th day after stroke demonstrated the construct and predictive validity of ASPOS. Moreover, it is more universal and can be used not only to assess the clinical condition, but also to reflect the functional status, reactivity and the degree of independence. Therefore, we hypothesize its particular usefulness for the proper and objective assessment of the subject’s condition during the diagnostic procedure in the emergency ward. Completing the questionnaire takes up to a minute and is a valuable supplement to gaps and deficiencies for certain clinical symptoms not included in the NIHSS, and fully reflects the actual and reliable stroke severity. Neurological examination of the patient is extended to include gait and swallowing assessments, which take a few more minutes, but provide information on key symptoms for posterior stroke, often overlooked in the initial diagnosis, which underestimates the patient’s condition and may delay or resign from extended vascular diagnosis. In our opinion, it is more appropriate to spend a few more minutes on a more reliable assessment of the patient, which will provide the basis for firm and quick diagnostic and therapeutic decisions than to observe the dynamics of symptoms for several minutes and wait for a significant deterioration, hesitating and delaying further treatment. In many cases only the neurological deterioration forces us to make faster decisions. Moreover, we hypothesize that additional time for a better assessment of stroke severity can ultimately shorten and accelerate diagnostic process and contribute to a faster initiation of reperfusion therapy.

Our device might be used not only for the reliable assessment of posterior circulation strokes. Due to its predictive properties, we hypothesized its usefulness as a criterion for undertaking the qualification activities for mechanical thrombectomy. Two-thirds of our patients obtained a total value of ASPOS that did not exceed two points, which was also the median score. This was not related to the approach, even with the generally accepted limit of six points using NIHSS. It was revealed that the probability of achieving this limit was significantly increased after obtaining three points by ASPOS; it increased successively with each subsequent point. In our opinion, this dependence can be utilized, and the value of three points by ASPOS can be considered as a border value for the initial selection of patients for extended diagnostic procedures in the direction of detecting occlusion of large intracranial vessels. However, we realized that, due to the lower accuracy of NIHSS in posterior stroke, assuming a value of six points may be doubtful, but there is currently no better point of reference.

It is not our intention to completely replace or abolish the NIHSS. When developing and presenting a new tool, we sought to draw attention to the possibility of its use as a realistic alternative or additional tool dedicated to posterior strokes for a better and more reliable measurement of specific symptoms compared to anterior stroke. ASPOS may support NIHSS in reducing and correcting the proven differences in prolonged diagnostic time and lower the quality of treatment in patients with posterior circulation stroke versus anterior circulation [20]. However, not even the best clinimetric scale can replace the knowledge and experience of a clinician.

The authors are aware that the current study has some limitations. Due to a moderate sample size, further multicenter research on larger cohorts is needed to confirm our findings. Our analysis did not cover the entire cross section of posterior circulation stroke subjects. For formal and bioethical reasons, a large group of severe stroke subjects was procedurally excluded. In the next step, validation of ASPOS should be performed among stroke patients who undergo reperfusion therapy. A limitation of the tool is that one of the items did not obtain a satisfactorily high discriminant value. However, the overall reliability of the device is sufficient, which, in view of the large variety of clinical manifestations in this group of strokes, is an advantage of the scale.

## 5. Conclusions

ASPOS is a novel, valid, and reliable diagnostic tool that can be used to assess posterior circulation strokes. It achieved appropriate psychometric features, high reproducibility, and easy performance, which makes it suitable for everyday practice in stroke units. It can contribute to a more comprehensive estimation of posterior stroke and, additionally, due to its predictive properties, it can be used to more accurately select candidates for specific treatment.

## Figures and Tables

**Figure 1 brainsci-11-00424-f001:**
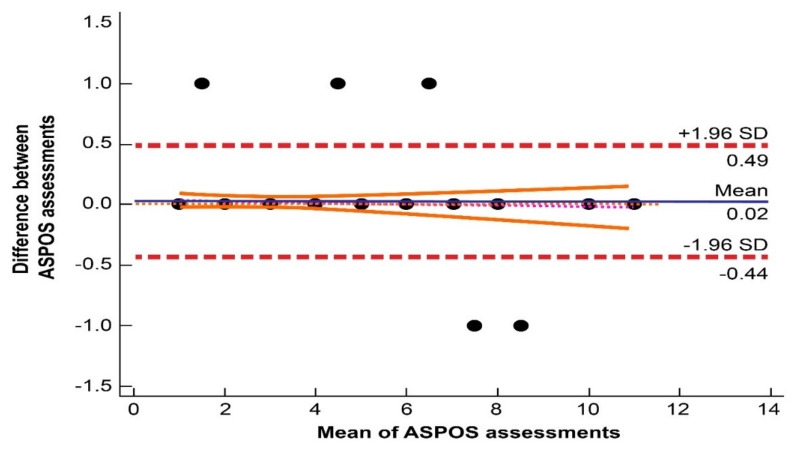
Bland–Altman diagram presenting the high repeatability of Adam’s Scale of Posterior Stroke (ASPOS).The distribution of points (black plots) is based on the mean and the difference from the total ASPOS score obtained by two randomly selected researchers. The area between the dashed red lines indicates the limits of agreement. The mean of the limits of agreement is presented as a blue line. The area between the solid orange lines represents the 95% confidence interval of the regression line (a line composed of red dots).

**Figure 2 brainsci-11-00424-f002:**
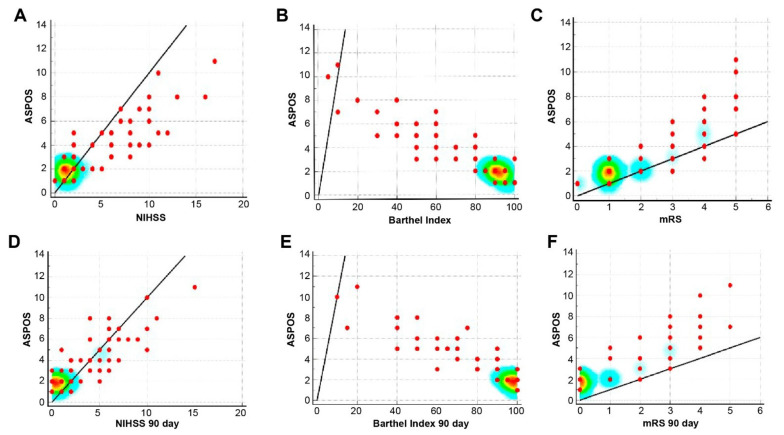
Construct (**A**–**C**) and predictive (**D**–**F**) validity of Adam’s Scale of Posterior Stroke (ASPOS).

**Figure 3 brainsci-11-00424-f003:**
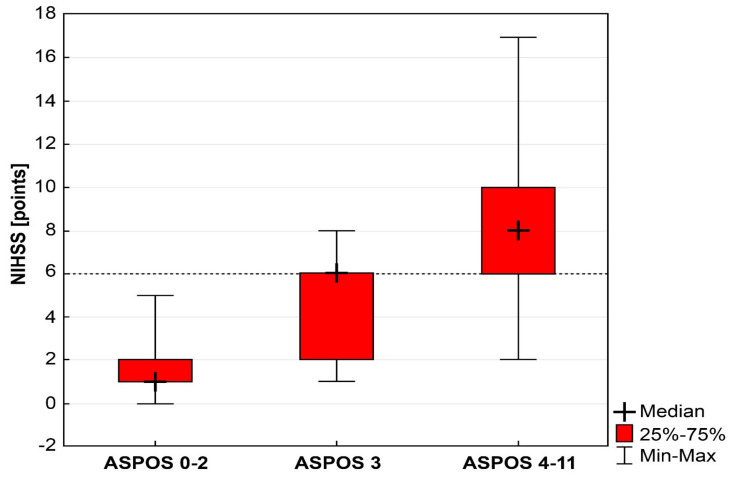
Predictive properties of Adam’s Scale of Posterior Stroke (ASPOS) in the assessment of posterior stroke severity in relation to the National Institutes of Health Stroke Scale (NIHSS) estimation. The dashed line marks the contractually accepted limit of stroke severity in the NIHSS (6 points), which is the criterion of qualification for a thrombectomy. A total ASPOS score of three points significantly increases the probability of reaching the defined limit, obtaining the median equal to six points in the NIHSS.

**Table 1 brainsci-11-00424-t001:** Adam’s Scale of Posterior Stroke (ASPOS).

Item	Score
**Reactivity**	0. conscious 1. somnolence, confusion 2. sopor 3. coma
**Eyes**	0. normal eye movement and visual fields 1. nystagmus, double vision, hemianopia 2. eye movement disturbances 3. oftalmoplegia, cortical blindness
Pharynx	0. normal swallowing, no dysarthria 1. mild dysarthria 2. moderate dysarthria, choking on liquids 3. anarthria, choking on solid foods, nosogastric tube
**Strength**	0. without motor deficit of limbs or face 1. mild motor deficit of limbs or face 2. moderate/severe motor deficit of limbs or face 3. limb paralysis
Balance	0. Romberg’s attempt negative, normal gait 1. guided walk, Romberg’s attempt unstable 2. walking with aids or help of another person 3. bedridden
**Ataxia**	0. without ataxia 1. ataxia present in one limb 2. ataxia present in two limbs
**Sensory**	0. without reactive and defective sensory deficit 1. paraesthesia, facial or single limb hypoaesthesia 2. hemianesthesia

Several items have been marked with a bold font to emphasize their similarity to items on the National Institutes of Health Stroke Scale.

**Table 2 brainsci-11-00424-t002:** Baseline characteristics of the participants (*n* = 126).

Parameter	Value
Age	69 (40–95)
Sex:	
Male	65 (51.6%)
Female	61 (48.4%)
Stroke etiology:	
Large vessel disease	7 (5.5%)
Small vessel disease	44 (34.9%)
Cardioembolism	42 (33.4%)
Not specified	33 (26.2%)
NIHSS on admission	2 (1–17)
mRS on admission	1 (0–5)
Barthel Index on admission	90 (5–100)
GCS on admission	15 (10–15)
Risk factors:	
Hypertension	102 (81%)
Diabetes	37 (29.4%)
Smoking	32 (25.4%)
Obesity	27 (21.4%)
BMI	27.65 (19.6–38.94)
Ischemic heart disease	26 (20.6%)
Hyperlipidemia	41 (32.5%)
Alcohol abuse	10 (7.9%)
Atrial fibrillation	42 (33.4%)

NIHSS—the National Institutes of Health Stroke Scale; mRS—modified Rankin Scale; BMI—Body Mass Index; GCS—Glasgow Coma Scale. The results of age, clinimetric scales and BMI are expressed as median and range, the results of sex, stroke etiology and risk factors—as N and percentage.

**Table 3 brainsci-11-00424-t003:** Selected psychometric properties of individual items in Adam’s Scale of Posterior Stroke (ASPOS).

Item	Correlation with Other Items (Discrimatory Power of Item)	Cronbach’s Alpha when Item was Removed
Reactivity	0.53	0.73
Eyes	0.02	0.79
Pharynx	0.66	0.66
Strength	0.51	0.70
Balance	0.58	0.68
Ataxia	0.57	0.69
Sensory	0.56	0.71

**Table 4 brainsci-11-00424-t004:** Inter-rater and intra-rater reliability of Adam’s Scale of Posterior Stroke (ASPOS).

Item	Inter-Rater Reliability		Intra-Rater Reliability			
	ICC	95% CI	ICC	95% CI	Weighted κ	95% CI
Reactivity	1.00	-	1.00	-	1.00	-
Eyes	0.97	0.96–0.98	0.97	0.95–0.98	0.96	0.91–1.00
Pharynx	0.97	0.96–0.98	0.97	0.96–0.98	0.94	0.87–1.00
Strength	0.98	0.98–0.99	0.97	0.96–0.98	0.96	0.90–1.00
Balance	0.98	0.97–0.98	0.97	0.96–0.98	0.96	0.90–1.00
Ataxia	0.99	0.98–0.99	0.98	0.97–0.99	0.97	0.93–1.00
Sensory	1.00	-	1.00	-	1.00	

ICC—intraclass correlation coefficient; CI—Confidence Interval; κ—Cohen’s kappa value.

## Data Availability

The data that support the findings of this study are available from the corresponding author upon request.

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
