# Peer review of "Introducing Adam’s Scale of Posterior Stroke (ASPOS): A Novel Validated Tool to Assess and Predict Posterior Circulation Strokes"

_brainsci, 2021, doi:10.3390/brainsci11040424_

Round 1

Reviewer 1 Report

The authors developed a new scoring system to assess symptoms know to be related to posterior circulation stroke. The system was termed ASPOS.

Although the reviewer agrees, that the most frequently used NIH stroke scale has limitations in the assessment of posterior circulation stroke, some methodological flaws exist, impeding publication of the proposed score in its current form.

1.: It is intransparent, how exactly the development process of the new score has been. From  a methological point of view, it would have been better to first comprehensively screen all patients with posterior circulation stroke, and then to derive the most frequent symptoms from this assessment to build a score on.

2. The reviewer has some doubts about the predictive performance. The predicitive performance (e.g. for performance of Mechanical thrombectomy) has been derived from the local monocentric clinical practice, and might differ in other hospitals. E.g. a NIHSS of >= is no longer the cut-off, given that patients with lower NIHSS but LVO tend to deteriorate rapidly, although randomized clinical trials are ongoing, to confirm these observations.

3. The clinical use of using another scale in addition to the NIHSS must be better pointed out. What is the actual utility of the ASPOS? Performing the NIHSS plus a sound neurological examination would provide the same base to build clinical decision on. The high correlation with NIHSS furthermore limits its use. 

Minor comments:

4. Table 1 should be removed from the Methods to the results

5. In Table 2, the items identical to the NIHSS should be highlighted

6. Numbers should be provided with 1 or max of 2 decimals only

7. Figure 2: In the Figure Key, the meaning of the colors should be explained Furthermore, the figure 2 looks "descaled". In Figure 1, decimal "." instead of "," should be used consistently.

8. Figure 3: Figure 3 simply underlines, that the additional use of ASPOS does not provide new clinical insights not obtained from using the NIHSS already. Clinical utility would arise, as if the NIHSS would not at all detect clinical worsening in posterior circulating stroke, but it does, as shown in Figure 3.

Author Response

Response to Reviewer 1 Comments

At the beginning, I would like to thank You for the careful review of our study and the constructive comments that have been used to organize all issues and improve the work.

Point 1:

The authors developed a new scoring system to assess symptoms know to be related to posterior circulation stroke. The system was termed ASPOS.

Although the reviewer agrees, that the most frequently used NIH stroke scale has limitations in the assessment of posterior circulation stroke, some methodological flaws exist, impeding publication of the proposed score in its current form.

It is intransparent, how exactly the development process of the new score has been. From  a methological point of view, it would have been better to first comprehensively screen all patients with posterior circulation stroke, and then to derive the most frequent symptoms from this assessment to build a score on.

Response:

We fully share the opinion of the reviewer that the process of creating the tool has not been fully described and requires more detail. First, we screened patients with posterior stroke hospitalized in our department in 2017-2018 to identify and characterize the most common neurological symptoms. Then we categorized individual symptoms and created initial items in different configurations, giving different scores on a scale from 0 to max. 3 points for each item. Ultimately, we created 6 different pre-versions with 6 to even 9 items with different scores. All versions were refilled during the study of each patient. After completing the study, each version was analyzed in terms of psychometrical parameters, reliability and constructive and predictive accuracy. The final version was the one that achieved the highest design and clinical properties.

According to the Reviewer recommendation we have expanded the description of development process of the tool in Methodology Section (Lines 102-111).

Point 2:

The reviewer has some doubts about the predictive performance. The predicitive performance (e.g. for performance of Mechanical thrombectomy) has been derived from the local monocentric clinical practice, and might differ in other hospitals. E.g. a NIHSS of >= is no longer the cut-off, given that patients with lower NIHSS but LVO tend to deteriorate rapidly, although randomized clinical trials are ongoing, to confirm these observations.

Response:

We used international guidelines and criteria for stroke subjects that may be qualified to mechanical thrombectomy (Powers, W.J.; Rabinstein, A.A.; Ackerson, T.; Adeoye, O.M.; Bambakdis, N.C.; Becker, K.; Biller, H.; Brown, M.; Demaerschalk, B.M.; Hoh, B.; et al. American Heart Association Stroke Council: 2018 Guidelines for the Early Management of Patients with Acute Ischemic Stroke: A Guideline for Healthcare Professionals From the American Heart Association/American Stroke Association. Stroke 2018, 49, e46–e110, doi:10.1161/STR.0000000000000158;  updated in 2019)

NIHSS score of ≥6 is one of the most important criterion located in the highest power

of recommendation (Class I) and is followed by all the Centers of Neurology in Poland,

and also by the majority of the world. This clinimetric cut-off point is an objective and

 generally accepted parameter based on randomized clinical trials. In our opinion, we

could not accept any other value as predictive, since such a value has been officially

set in many international recommendations and current clinical trials. In our study,

we wanted to use the best reference point and our choice could not have been different.

We fully agree with the Reviewer that the prediction values cannot be based solely on

the NIHSS and other features such as the dynamics of symptoms or clinical deterioration

should be also taken into account. However, there is still a lack of objective and reliable

 benchmarks and cut-off points to relate to. Of course, when there are new approved

indications or reference points, we will adjust our results to subsequent variables in

 the next stages of research on the usefulness of ASPOS.

Point 3:

The clinical use of using another scale in addition to the NIHSS must be better pointed out. What is the actual utility of the ASPOS? Performing the NIHSS plus a sound neurological examination would provide the same base to build clinical decision on. The high correlation with NIHSS furthermore limits its use. 

Response:

One of the features of the validated scale is its construct validity. The only objective and required method to evaluate its accuracy is correlation with other recognized scales, such as the NIHSS. There is no other way to objectively judge whether our new tool correctly assess the severity of a stroke. Therefore, we believe that our scale would not be useful if it did not correlate with the NIHSS and each expert in the field of neurology would abandon its use as it would produce different results than the NIHSS.

The assessment of clinical utility of our tool requires further research on larger cohort of subjects in multi-center, international investigation. Our goal was to present a new tool, which obtained satisfactory parameters in the preliminary analysis, allowing it to be admitted to further stages. Only many years of verification will either confirm or deny its clinical usefulness.

The main aim of our study is to show that despite the diversity of the clinical manifestation of posterior strokes, it is possible to construct a tool that will consolidate all of them and at the same time covering all the most important aspects of this type of stroke.

Moreover, our device has also some advantages over the NIHSS:

1) It was designed according to the requirements of the International Quality of Life Assessment Project,

2) The overall internal consistency and homogeneity is better than the NIHSS (the validation of the NIHSS in aspect of psychometric values will be published soon in Plos One, where we show moderate scalability of the NIHSS),

3) It covers all the most important clinical symptoms of posterior stroke and does not overlook any significant symptoms,

4) it is more universal and can be used not only to assess the clinical state but also to reflect the functional status, reactivity and the degree of independence.

We pointed out all these advantages in Discussion Section ( Lines 216-219, 241-243).

We hypothesize that ASPOS may be used as alternative for the NIHSS or may support it in more reliable, objective and valid assessment of posterior stroke severity (reducing the limitations of the NIHSS in this aspect) and in many situations contribute to make key diagnostic and therapeutic decisions (Lines 257-263).

Point 4:

Table 1 should be removed from the Methods to the results.

Response:

Thank You for this suggestion. We have transfered it to the Results Section (as Table 2).

Point 5:

In Table 2, the items identical to the NIHSS should be highlighted.

Response:

As suggested by the Reviewer we have highlighted the same items.

Point 6:

Numbers should be provided with 1 or max of 2 decimals only.

Response;

According to the Reviewer recommendation we have shorten our results in the manuscript and in tables and provided them with two decimals. We only left the results that indicate high statistical significance (e.g. p<0.0001)

Point 7:

 Figure 2: In the Figure Key, the meaning of the colors should be explained Furthermore, the figure 2 looks "descaled". In Figure 1, decimal "." instead of "," should be used consistently.

Response:

We appreciate this comment. We have made the changes proposed by the Reviewer- we have explained the colors meaning, changed the punctuations and resized the Figure 2.

Point 8:

Figure 3: Figure 3 simply underlines, that the additional use of ASPOS does not provide new clinical insights not obtained from using the NIHSS already. Clinical utility would arise, as if the NIHSS would not at all detect clinical worsening in posterior circulating stroke, but it does, as shown in Figure 3.

Response:

As we mentioned in the Response 3 the high correlation with the NIHSS was required to obtain construct and predictive validity of the ASPOS. Without having other objective reference points, we cannot otherwise define the advantages of our new tool.

Additionally, we have made extensive changes of language and structure in our manuscript,  using the MPDI professional English editing service, to improve the style and grammar.

Thank You very much for all criticism. We hope that major revision of our paper made according the Reviewer’s guidelines will improve our manuscript.

Reviewer 2 Report

Author required to improve data presentation.

  1. What does that mean green bar, yellow, red dot lines on fig 1.
  2. What dose that mean color highlight on fig 2.
  3. Poor quality of graph. do not use same format of graph.

Author Response

Response to Reviewer 2 Comments

At the beginning, I would like to thank You for the careful review of our study and the constructive comments that have been used to organize all issues and improve the work.

Point 1:

Author required to improve data presentation.

  1. What does that mean green bar, yellow, red dot lines on fig 1.
  2. What dose that mean color highlight on fig 2.
  3. Poor quality of graph. do not use same format of graph

Response:

We are grateful to the Reviewer for this important remark. Therefore, we have extended the figure legends and captions and described the additional information to help the readers better understand the meaning of the figures (Lines 158-162, 181-183). We fully agree with the Reviewer that our data presentation was poor, so we have decided to  improve it in the best possible way and we asked the Editage- Artwork Professional Formatting Service to help us to make the corrections of our Figures. We have made every effort to improve data presentation to make it more transparent, clear and understood. The figures were resized and achieved higher resolution. We believed that revised Figures and extended captions will be acceptable by the Reviewer.

Thank You very much for all criticism.

We hope that revision of our paper made according the Reviewer’s guidelines

will improve our manuscript.

Reviewer 3 Report

Thank you for your thoughtful contribution.  I do think your article shows that the ASPOS is a reliable scoring system and I appreciate your correlation to the NIHSS cut off of 6.

My concerns for clinical applications are: low number of patients, excluded patients in comatose state (reactivity score limited by 2 points), excluded thrombectomy and it is not clear weather or not hemorrhagic stroke pathology was included and if so how it correlates to the ICH score.

I am hopeful future papers will look at the above concerns.

Author Response

Response to Reviewer 3 Comments

At the beginning, I would like to thank You for the careful review of our study and the constructive comments that have been used to organize all issues and improve the work.

Point 1:

Thank you for your thoughtful contribution.  I do think your article shows that the ASPOS is a reliable scoring system and I appreciate your correlation to the NIHSS cut off of 6.

 My concerns for clinical applications are: low number of patients, excluded patients in comatose state (reactivity score limited by 2 points), excluded thrombectomy and it is not clear weather or not hemorrhagic stroke pathology was included and if so how it correlates to the ICH score.

 I am hopeful future papers will look at the above concerns.

Response:

We are grateful to the Reviewer for this important and kind comments.

The main goal of our study was to present a novel tool and show its good psychometric properties and clinical accuracy.  As the Reviewer suggested the clinical utility of our tool will be verified and confirm in future research, based on large cohort, multi-center, international investigations.

In the next stages we will look at the concerns highlighted by the Reviewer, including in particular stroke subjects with extremely poor clinical condition and hemorrhagic strokes.

Thank You very much for all criticism.

Round 2

Reviewer 1 Report

The authors have mostly responded well to the reviewers concerns.
Some minor remarks remain:

Minor comments
(1) (New) Table 1 (ASPOS): Provide a table key, explaining the meaning of "bold" printed (NIHSS identical items). Reactivity should be printed in bold as well (as it equals to NIHSS LOC, 1a).

Discussion

(2) The authors are encouraged to include a more clear description of the "place" within the diagnostics algorithms and clinical workup of stroke patients, where ASPOS should be used to provide clinical utility in addition to existing tools.

(3) The reviewer does not understand, what is meant by "prolonged diagnostic time" in the following sentence found in the discussion:   "ASPOS may support NIHSS in reducing and correcting the proven differences in prolonged diagnostic time and lower the quality of treatment in patients with posterior circulation stroke versus anterior circulation [20]."

3a: Do the authors expect that diagnostic time shortens, if another scale/score is applied in addition to the NIHSS?

3b: The ASPOS includes time-consuming items that require swallowing diagnostics (Item "Pharynx"). Please include a brief statement about the time it takes to perform the ASPOS, and its resulting limitations for emergency situations.

Author Response

Response to Reviewer 1 Comments

At the beginning, I would like to thank You for the careful review of our study and the constructive comments that have been used to organize all issues and improve the work.

Point 1:

(New) Table 1 (ASPOS): Provide a table key, explaining the meaning of "bold" printed (NIHSS identical items). Reactivity should be printed in bold as well (as it equals to NIHSS LOC, 1a).

Response:

Thank You for this suggestion. We have added a table footnote, where we have indicated the information that the items in bold are similar to those used on the NIHSS scale. In line with the Reviewer’s recommendation we have also changed the font of “reactivity” to bold (Table 1).

Point 2:

The authors are encouraged to include a more clear description of the "place" within the diagnostics algorithms and clinical workup of stroke patients, where ASPOS should be used to provide clinical utility in addition to existing tools.

Response:

Thank Your for this remark. We have extended our Discussion Section to provide information about „place” of ASPOS in diagnostic algorithm, in particular in assessment of stroke severity in emergency ward for a more reliable and accurate evaluation of clinical condition and proper selection of candidates to mechanical thrombectomy (Lines 243-249, 261-263, 295-297).

Point 3:

The reviewer does not understand, what is meant by "prolonged diagnostic time" in the following sentence found in the discussion:   "ASPOS may support NIHSS in reducing and correcting the proven differences in prolonged diagnostic time and lower the quality of treatment in patients with posterior circulation stroke versus anterior circulation [20]."

Response:

In the citing paper the authors conclude that diagnostic time in emergency ward for posterior strokes is much prolonged (almost twenty-thirty minutes) compared to anterior strokes, that is the main cause of delayed treatment and contributes to worse outcome. One of the reasons is imperfect clinimetric tools dedicated to posterior strokes. Therefore, we provide ASPOS that might support the NIHSS in a better measurement of stroke severity and reduce the diagnostic time.

Point 4:

Do the authors expect that diagnostic time shortens, if another scale/score is applied in addition to the NIHSS?

Response:

In our opinion, it is more appropriate to spend a little more minutes on a more reliable assessment of the patient, which will provide the basis for firm and quick diagnostic and therapeutic decisions than to observe the patient, the dynamics of symptoms for several minutes and wait for a significant deterioration, hesitating and delaying further treatment. We know from our own experience that in many cases only the deterioration of the neurological condition forces us to make faster decisions.

We included our opinion in Discussion Section (Lines 253-258).

Point 5:

The ASPOS includes time-consuming items that require swallowing diagnostics (Item "Pharynx"). Please include a brief statement about the time it takes to perform the ASPOS, and its resulting limitations for emergency situations.

Response:

As suggested by the Reviewer we included such a statement about the time to perform ASPOS (Lines 247-253). However in our opinion that additional time for a better assessment of stroke severity can ultimately shorten and accelerate diagnostic decision making and contribute to a faster initiation of reperfusion therapy (Lines 258-260).

We believe that above revisions will significantly improve our paper.

Thank You once again for valuable comments and suggestions.